# Antiviral Treatment of Maternal and Congenital Cytomegalovirus (CMV) Infections

**DOI:** 10.3390/v15102116

**Published:** 2023-10-19

**Authors:** Swetha Pinninti, Suresh Boppana

**Affiliations:** 1Department of Pediatrics, Heersink School of Medicine, University of Alabama at Birmingham, Birmingham, AL 35233, USA; sbboppana@uabmc.edu; 2Department of Microbiology, Heersink School of Medicine, University of Alabama at Birmingham, Birmingham, AL 35233, USA

**Keywords:** cytomegalovirus, congenital, ganciclovir, valacyclovir, maternal–fetal infections

## Abstract

Human Cytomegalovirus (HCMV) is a ubiquitous member of the Herpesviridae family, responsible for the most common congenital viral infection—congenital Cytomegalovirus (cCMV) infection. While a majority of HCMV infections in children and adults are asymptomatic, HCMV is well known to cause severe infections in the immunocompromised individual and maternal infections with variable long-term sequelae after maternal–fetal transmission with primary or nonprimary infections. HCMV seroprevalence and cCMV incidence vary by geographic area and demographic characteristics like race and socioeconomic status. While cCMV birth prevalence ranges from 0.2% to 6% in different parts of the world, it is influenced by regional HCMV seroprevalence rates. HCMV screening during pregnancy is not routinely offered due to lack of awareness, hurdles to accurate diagnosis, and lack of well-established effective treatment options during pregnancy. This review will focus on antiviral treatment options currently available for use during pregnancy and in the newborn period for the treatment of maternal and congenital HCMV infections.

## 1. Treatment of Maternal Human Cytomegalovirus (HCMV) Infections

### 1.1. Introduction

Congenital CMV (cCMV)-associated sequelae can occur in infections following both primary and nonprimary maternal HCMV infections. Primary maternal HCMV infections are defined as the acquisition of HCMV during pregnancy in a person without pre-existing HCMV-specific IgG antibodies. Non-primary maternal HCMV infections are defined as active HCMV infections in pregnant individuals with pre-existing HCMV-IgG antibodies. The majority of cCMV infections, especially in populations with high seroprevalence, occur following non-primary maternal infections secondary to reactivation of endogenous HCMV strains or reinfections with a different virus strain [1,2,3]. Among women with primary HCMV infections during pregnancy, although the risk of intrauterine transmission of HCMV increases with gestational age, severe cCMV-associated sequelae occur most often in early gestation infections [3,4,5].

Prenatal HCMV screening is not routinely offered because of the lack of effective interventions to prevent in utero transmission of HCMV and HCMV-associated sequelae besides the inability to identify individuals with non-primary maternal infections who are at increased risk for intrauterine transmission of HCMV.

Several intervention strategies aimed at preventing maternal–fetal transmission in primary HCMV infections have been studied, including behavioral risk reduction measures, HCMV hyperimmune globulin (HIG), and antiviral therapies. Due to the complexities of definitively diagnosing non-primary HCMV infections and the lack of tools to identify those with non-primary infection at increased risk for intrauterine transmission, interventions to prevent maternal–fetal HCMV transmission in this scenario are nonexistent.

### 1.2. Behavioral and Hygiene Intervention Strategies

Increasing the awareness of HCMV using educational and behavioral interventions to decrease exposure to HCMV during pregnancy has been met with variable success rates. A mixed interventional and observational controlled study by Revello et al. in seronegative women showed that educational and hygiene measures significantly reduced primary maternal HCMV infection, thereby reducing cCMV infection [6]. In this study, the seroconversion rate in seronegative patients who received specific hygiene information was 1.2%, compared to 7.6% in patients who were not screened for HCMV or provided hygiene measures. Although this study only included seronegative women, this strategy may also be effective in women with non-primary infection during pregnancy, secondary to reinfection with a different virus strain. It is likely that hygiene education for all women in early pregnancy and those trying to conceive may reduce both primary and non-primary maternal HCMV infections. It is interesting that risk reduction education did not affect seroconversion rates in nonpregnant women [7].

### 1.3. HCMV Hyperimmune Globulin (HIG)

The use of HIG was considered a promising intervention for the prevention of maternal–fetal transmission in primary infections based on open-label and observational cohort studies [8,9,10]. However, a randomized, placebo-controlled trial that compared the efficacy of HIG infusions with a placebo did not show a significant decrease in rates of fetal infection (30% vs. 44%; *p* = 0.13) [11]. Another large randomized, controlled trial by the NICHD MFMU Network examined the efficacy of monthly HIG infusions compared to placebo in 394 pregnant people with primary HCMV infection at <24 weeks gestation and did not demonstrate a significant difference in fetal transmission between the treatment arms (RR—1.17; *p* = 0 .42). Additionally, while not statistically significant, rates of preterm birth and newborns with birth weights < 5th percentile were higher in women who received HIG compared to those who received placebo, as well as side effects like headache and shaking chills [12]. Therefore, HIG is not recommended as an intervention option for the prevention of maternal–fetal HCMV transmission in primary HCMV infections. Although observational studies using biweekly HCMV HIG during the first trimester provided promising results, these findings should be confirmed in randomized studies [13,14].

### 1.4. Antiviral Treatment

Recent studies have shown the effectiveness of valacyclovir treatment in primary maternal HCMV infections to prevent fetal HCMV infection, as discussed below. The premise of this intervention is that a decrease in HCMV viral load (VL) leads to a reduction of vertical transmission of HCMV during pregnancy and sequelae in newborns. While antiviral treatment with valganciclovir is currently only reserved for infants with symptomatic cCMV to improve hearing and neurodevelopmental outcomes, due to associated toxicities, secondary and tertiary interventions during pregnancy using this agent are not currently recommended. Moreover, valacyclovir has anti-HCMV activity and has been shown to be effective in preventing HCMV reactivation in renal transplant recipients [15] and in smaller cohort studies to have adequate placental transfer of the drug and an acceptable tolerance and safety profile in pregnant women.

One randomized clinical trial and several observational cohort studies have been performed to determine the efficacy of oral valacyclovir in preventing maternal–fetal HCMV transmission in primary HCMV infections. In the only prospective, randomized, double-blind, placebo-controlled trial by Shahar-Nissan et al. from Israel, 90 pregnant women with serological evidence of primary HCMV infection acquired in the periconceptional period or during the first trimester were randomized to receive either oral valacyclovir (8 g/day twice daily) or placebo, with the primary objective to determine the rate of vertical transmission of HCMV [16]. Of 90 women, 11% from the valacyclovir group, compared to 30% from the placebo arm, tested positive for HCMV on amniocentesis at 21–22 weeks gestation (*p* = 0.27; OR—0.29, CI: 0.09–0.9) after a median 6 weeks of treatment. No significant difference between HCMV positivity on amniocentesis was observed between treatment and placebo groups for periconceptionally acquired primary HCMV infection (12% vs. 13%, respectively; *p* = 0.91). However, among those with first-trimester infections, HCMV-positive amniocentesis was significantly less likely in the valacyclovir group compared to the placebo (11% vs. 48%, respectively; *p* = 0.02). Of note, women with first-trimester infections in the treatment arm initiated the treatment earlier than those with periconceptional infections, suggesting a role for the timing of treatment initiation in these circumstances (43.84 days vs. 60.58 days; *p* = 0.0026). The drug was overall well tolerated with minimal side effects (thrombocytopenia, nausea, headache, and abdominal pain).

When fetal involvement/cCMV was assessed as a secondary outcome in this study, although participants in the treatment group had an overall lower incidence of cCMV-associated morbidity (hearing loss, CNS imaging findings) compared to those who received the placebo (7% in the treatment group vs. 16% in placebo; OR—0.38; CI 0.09–1.56), this difference was not statistically significant. Six newborns were diagnosed with cCMV among women with negative HCMV testing from amniocentesis (four in the treatment group vs. two in placebo), with five out of six with asymptomatic cCMV. The investigators propose delayed fetal infection after cessation of treatment compounded by increased transmission with advanced gestational age as the likely cause for these cases. While the results of this study are encouraging, the study is limited by the small sample size, lack of universal prenatal HCMV screening, and because the majority of cCMV infections occur following non-primary maternal infection. Moreover, the ideal duration of treatment, cost effectiveness, and the effect of timing of initiation of treatment were not addressed as part of this study.

Subsequently, in a case-control study with propensity score matching by Faure-Bardon et al., the utility of HCMV serology screening in the first trimester, followed by secondary HCMV prevention with valacyclovir, was pursued [17]. The primary outcome assessed in this study was fetal HCMV infection based on amniotic fluid HCMV DNA PCR testing at 17–22 weeks gestation. Of 310 pregnant women with primary HCMV infection, 65 women each were enrolled into two cohorts: receiving valacyclovir treatment or as controls. In the treatment group, valacyclovir was initiated at a median gestational age of 12.71 weeks (IQR: 10–13.86 weeks) and treated for a median of 35 days (IQR: 26–54 days). The rate of maternal to fetal HCMV transmission, based on amniocentesis, was found to be lower in women who were treated with valacyclovir compared to untreated controls (12% vs. 29%, respectively; *p* = 0.029) for the entire cohort, significantly so for first-trimester infections (*p* = 0.027) compared to periconceptional infections (*p* = 0.6). The authors postulate that the increased duration between infection and testing and the initiation of valacyclovir in women with periconceptional infection is likely responsible for the decreased efficacy of the antiviral treatment in these infections. Notably, one participant developed acute renal failure four weeks after the initiation of treatment but resolved spontaneously after discontinuation of antiviral treatment. While this study documents the overall acceptability and safety profile of valacyclovir during pregnancy, the lack of newborn testing for cCMV, a potential for missing late transmissions or infections after amniocentesis, and the case-control study model prevent the generalization of these findings.

Similar data have been replicated and reported from large multicenter observational or cohort studies. In a retrospective, multicenter study by Egloff et al., the rate of maternal–fetal transmission of HCMV was compared in women with periconceptional or first-trimester primary infections between women who received oral valacyclovir with a control group of women who did not receive any intervention [18]. The primary aim of the study was to determine the efficacy of valacyclovir treatment beyond the first trimester for the secondary prevention of cCMV. The efficacy of the valacyclovir treatment was assessed with logistic regression analysis adjusted for a propensity score and showed that valacyclovir treatment was significantly associated with an overall reduction in the rate of maternal–fetal HCMV transmission (OR—0.40 (95% CI, 0.18–0.90); *p* = 0.029), with no statistical significance achieved for periconceptional or second-trimester infections. Among the women who were treated, the rate of HCMV transmission was higher in the presence of maternal viremia, with a trend towards greater efficacy of valacyclovir when viremia was detected compared to those without viremia. In addition, treated pregnant women had lower HCMV viral load levels in the amniotic fluid (*p* = 0.44) and significantly lower newborn cord blood viral load levels (*p* = 0.02). Valacyclovir was overall well tolerated, except for reversible renal failure in one participant. While this is the largest cohort to date to assess the efficacy of antiviral treatment for the secondary prevention of cCMV, the retrospective nature of the study is a huge limitation.

Similarly, results from a recently published Italian multicenter observational retro-prospective study (MEGAL-ITALI) reported data from 447 pregnant women [19]. In this study, 205 (45.8%) were allocated to the treatment group with oral valacyclovir and showed a significant reduction in rates of HCMV positivity on amniocentesis (OR: 0.39; CI: 0.22–0.68; *p* = 0.005). Similar to other reported data, the mean duration between the diagnosis of primary infection and initiation of treatment was 7.9 weeks, with a mean duration of treatment of 57.8 days. Valacyclovir was well tolerated by participants, with renal toxicity reported in only 1.9% of women. Similar to other observational studies discussed above, the observational, retrospective, and prospective nature of this trial, with a lack of systematic HCMV screening during pregnancy, limits the generalization of the results of this study.

Chatzakis performed an individual patient data meta-analysis that included the above-mentioned studies with a primary endpoint of amniocentesis results [20]. In this meta-analysis, 527 women were included from three trials—218 (42.3%) in the treatment arm and 309 (58.6%) in the placebo/no intervention arm. The mean maternal age was 32.2 years, with a mean gestational age at treatment initiation of 11.4 gestational weeks and a mean duration between seroconversion to treatment initiation of 8.4 weeks. Participants were treated for a mean duration of 60.3 days. Of 515 women who underwent amniocentesis, 24/217 (11.1%) vs. 76/298 (25.5%) had a positive amniocentesis for HCMV DNA PCR in the treated vs. untreated cohorts (*p* < 0.001). Similarly, to determine secondary outcomes, data from 396 neonates were available, with a cCMV incidence of 41.1% in the placebo group vs. 19.2% in the valacyclovir group. Gestational age at initiation of treatment was a significant factor, with earlier initiation associated with a higher likelihood of transmission. On a safety assessment available from 139 participants, 20% reported nausea and headache, while 2.1% developed acute kidney injury.

Despite the promising results from these studies, the findings of the efficacy of antiviral therapy during pregnancy are limited by the fact that the studies are only applicable to primary maternal infections and would require the widespread application of universal prenatal HCMV screening. In addition, a systematic review and meta-analysis showed that on GRADE assessment, the quality of evidence that valacyclovir reduces the risk of cCMV was very low [21]. Moreover, based on current estimates and HCMV seroprevalence and primary infection rates, universal first-trimester HCMV screening with valacyclovir prophylaxis is not believed to be a cost-effective strategy [22].

### 1.5. Treatment of Fetal CMV Infection

The barriers to detection and treatment of fetal HCMV infections are similar to those associated with maternal HCMV primary infections—a lack of routine screening during pregnancy, a small number of primary HCMV infections overall, and a lack of reliable diagnostic methodology for nonprimary infections. While most of the burden attributable to cCMV is due to nonprimary HCMV infections, the risk of transplacental HCMV transmission is higher with primary infections, ranging from 21% in the periconceptional period to >50% in the third trimester, albeit with lower severity of cCMV [5]. However, multiple reports have documented severe infections and long-term sequelae with second- and third-trimester infections [23,24,25]. While the goal of using HIG and oral valacyclovir for primary HCMV infections aims to decrease maternal to fetal HCMV transmission and, consequently, the incidence of cCMV, studies have documented some decreases in cCMV severity too. Of note, pregnancies with moderate-to-severe fetal disease or CNS involvement have been excluded from treatment trials due to the low likelihood of improving the outcome. In a recently published systematic review and meta-analysis that included 620 women overall from eight studies, women treated with valacyclovir had a higher incidence of asymptomatic cCMV based on pooled data from 132 fetuses (OR—2.98; *p* = 0.021) without a significant effect on symptoms in the newborn period (*p*  =  0.092), suggesting that once fetal infection is established, prenatal valacyclovir therapy is unlikely to improve prognosis [21].

### 1.6. HCMV Vaccines

No HCMV vaccine is currently licensed to prevent maternal–fetal HCMV transmission. However, multiple HCMV vaccines are in various stages of development, and early clinical trials with licensure are likely in the near future [26]. Candidate vaccines that are being evaluated include those targeting envelope glycoproteins, glycoprotein B, the pentameric glycoprotein complex (gH,gL,Ul128,Ul130-131), and candidate vaccines that express glycoprotein B with nonenvelope virion proteins and nonstructural proteins, with a focus on inducing antiviral antibodies and robust CD4/CD* + T lymphocyte responses. Previously, two vaccine trials that utilized recombinant glycoprotein B vaccine in HCMV-seronegative women only showed an efficacy of 45–50% for HCMV seroconversion, with waning immunity over time [27,28]. In the recent past, an mRNA-based HCMV vaccine that codes for the pentamer complex and glycoprotein B successfully completed a phase 2 safety and immunogenicity trial and is actively enrolling healthy participants 16–40 years in a phase 3, randomized, placebo-controlled trial (NCT05085366).

## 2. Treatment of Congenital CMV (cCMV)

### 2.1. Identification of Infants with Congenital CMV Infection

The high levels of HCMV DNA in the saliva and urine of newborns enable the use of either specimen to identify congenitally infected newborns [29,30,31,32,33,34]. Irrespective of the kind of specimen utilized for diagnosis, samples should be collected from the infant within 2–3 weeks of life to make a definitive diagnosis of cCMV and differentiate from postnatally acquired HCMV infections.

### 2.2. Newborn HCMV Screening

Most (90%) HCMV-infected newborns are not identified in the newborn period due to a lack of symptoms (asymptomatic infection). Among children with symptomatic cCMV, 50–60% are at risk of developing long-term sequelae—SNHL is the commonest, followed by developmental disabilities, chorioretinitis, and cerebral palsy [35,36]. Among children with asymptomatic cCMV, 10–15% will develop SNHL [37]. In children with cCMV, 33–50% of SNHL develops beyond the newborn period (late onset). In those with SNHL, 50% will continue to have further deterioration (progressive loss), frequently in children with symptomatic cCMV. A characteristic feature of HCMV-related hearing loss is fluctuating hearing loss that may occur unilaterally or bilaterally. Overall, SNHL in cCMV is variable with respect to time of onset, laterality, progression, and fluctuance [38]. Increasingly, universal newborn HCMV screening is being proposed for early identification of all infants with cCMV so that children with SNHL and other sequelae can be identified early to provide intervention and improve speech and language outcomes. In June 2021, Minnesota became the first state to enact universal newborn HCMV screening, and multiple other states currently require education of pregnant women, the public, and professionals about cCMV. Another strategy that has been utilized is implementing targeted HCMV screening, wherein HCMV screening is performed when an infant fails the newborn hearing screening. However, with this strategy, it is estimated only 57% of HCMV-infected infants with confirmed SNHL at birth would be identified, and it will miss children in whom SNHL is late onset and progressive in later childhood [39].

### 2.3. Predictors of Outcome in cCMV

Multiple studies have explored utilizing prenatal and neonatal findings as biomarkers for early identification of cCMV sequelae, with limited success. CNS involvement, symptomatic cCMV, high HCMV viral load at birth, and failed newborn hearing screen have been variably identified as biomarkers for long-term sequelae in children with cCMV [40,41]. Studies have documented that compared to those with symptomatic cCMV, SNHL in children with asymptomatic cCMV may not adversely impact quality of life, based on studies of longitudinal follow-up [42]. Studies that evaluated abnormalities detected on fetal and neonatal cranial imaging (ultrasound and MRI) as prognostic indicators have been inconclusive to determine a definitive association between the presence of imaging abnormalities, symptomatic disease, and hearing loss [43,44]. Most of these studies, however, are limited by small cohort sizes and the inclusion of predominantly symptomatic children with cCMV, limiting the definitive identification of newborns with cCMV at risk of developing sequelae or those that would benefit from antiviral treatment.

### 2.4. Treatment of the Newborn

In the context of cCMV, the National Institutes of Health (NIH)-funded Collaborative Antiviral Study Group (CASG) spearheaded most studies conducted to date for the treatment of cCMV-associated hearing and developmental outcomes. The dose-determining phase II studies for ganciclovir for neonates with cCMV with CNS involvement were successfully conducted in the 1980′s [45,46,47], followed by a randomized, placebo-controlled, phase III study to assess the effect of intravenous ganciclovir therapy on hearing in symptomatic cCMV with CNS involvement [48].

Of the 100 children enrolled in the study, evaluable data were only available for 42 children who underwent hearing evaluations at enrollment and at 6 months. Ganciclovir recipients, compared to children in the control group, showed some, but not statistically significant, improvement in or maintained hearing status (84% vs. 59%, respectively; *p* = 0.6) at 6 months. Additionally, none of the children included in the ganciclovir group had worsening hearing status compared to the placebo group (*p* < 0.01). In a subgroup of children with follow-up to 1 year and beyond, fewer ganciclovir-treated patients had hearing deterioration. However, two-thirds of ganciclovir recipients developed grade 3 to 4 neutropenia during treatment (63% vs. 21% controls; *p* < 0.01). Despite these promising results, due to the loss to the follow-up of more than half this cohort, concern remained for follow-up bias that could have influenced the conclusions of this trial. However, given the substantial side-effect profile, the need for IV access and close follow up during treatment, and unknown long-term effects (carcinogenicity and gonadal toxicity in animal models), ganciclovir is recommended/offered as a treatment option only for newborns with symptomatic cCMV with CNS involvement. In a follow-up study that compared the neurodevelopmental outcome of ganciclovir treatment in this cohort (developmental testing at 6 weeks, 6 months, and 1 year), treated infants were shown to have fewer developmental delays [49].

Follow-up CASG studies determined the oral dose of valganciclovir equivalent to the intravenous dose of ganciclovir, standardizing the treatment of symptomatic congenital CMV with CNS manifestations with 6 weeks of either IV ganciclovir or oral valganciclovir [50]. In a randomized, placebo-controlled trial, 96 neonates were randomized to receive 6 months vs. 6 weeks of oral valganciclovir and followed for 2 years for hearing and neurodevelopmental outcomes [51]. Despite a lack of improved audiological outcomes at 6 months between the groups (*p* = 0.41), at 12 months, more children treated for 6 months had hearing improvement than those treated for only 6 weeks (73% vs. 57%, *p* = 0.01). The benefit was sustained at the 24-month testing (77% vs. 64%, *p* = 0.04), including neurodevelopmental scores (*p* = 0.004). Grade 3–4 neutropenia was reported in 19% of the cohort, predominantly during the first 6 weeks, albeit with no significant differences between groups treated for 6 weeks or 6 months (*p* = 0.64). The international consensus includes recommendations for the treatment of cCMV and recommends treating newborns with moderate–severe symptomatic cCMV with or without CNS involvement for 6 months with valganciclovir to improve audiological and developmental outcomes with close follow-up [52]. Treatment initiation beyond the first month of life or treatment of isolated SNHL is not currently recommended due to a lack of data suggesting benefit. Despite the lack of recommendations for treatment beyond these parameters, data from a multicenter electronic health record dataset that evaluated ganciclovir and valganciclovir use among infants with cCMV showed that 50% of infants with cCMV were started on an antiviral, with the majority of valganciclovir prescriptions being initiated beyond the neonatal period [53]. Due to the lack of long-term benefit and safety data for the antiviral treatment options for cCMV, the authors do not currently endorse treatment indications outside of the currently recommended parameters. Randomized clinical trials with evidence for antiviral treatment efficacy in HCMV maternal and congenital infections are highlighted in Table 1.

Outside of the treatment parameters listed, few studies have explored the utility of antiviral treatment in cCMV in children with asymptomatic cCMV, isolated SNHL, treatment initiation beyond the neonatal period, as well as the long-term benefit and toxicity. Pasternak et al. reported data from 59 children with isolated SNHL (65% with unilateral SNHL) treated with valganciclovir initiated within 12 weeks of birth and treated for 12 months. Ears with milder hearing loss were more likely to improve (92.6% vs. 70% vs. 15.7%, respectively, in those with mild, moderate, and severe SNHL; *p* < 0.001) [54]. The main side effect observed in the study was transient neutropenia (32% of the cohort). Subsequently, in a small series of 16 children with symptomatic cCMV who received valganciclovir for one year and were then followed for an average of 3.2 years, a measurable worsening of hearing function over time was documented, suggesting that the improvement with antiviral treatment is likely temporary [55]. Similarly, in a longitudinal study of 76 children with symptomatic cCMV who were followed for hearing outcomes through adolescence, no significant differences in the frequency of severe hearing loss were documented in children treated with ganciclovir vs. those who were untreated [56]. Additional clinical trials are currently underway that might provide data in the near future to address these concerns (NCT02005822, NCT03107871). However, a phase II open-label trial to evaluate valganciclovir as a treatment to prevent the development of SNHL in infants with asymptomatic cCMV was recently suspended due to safety concerns (NCT03301415).

To date, ganciclovir/valganciclovir resistance has only occasionally been reported in children with cCMV undergoing treatment [57]. However, taking into consideration the lack of data on the toxicity and long-term benefits, in addition to the safety monitoring and compliance needed in the short term, it is important for clinicians to be cautious about treating infants with cCMV with antiviral therapies outside of the currently recommended parameters.

### 2.5. Recommended Long-Term Follow-Up

No standard guidelines exist for the long-term follow-up of children with cCMV. In 2017, consensus recommendations were published to provide guidance and recommendations for audiological, ophthalmological, and neurodevelopmental follow-up of children with cCMV [52]. The current recommendation is for children with cCMV to undergo serial audiological assessments, beginning in the newborn period, to be repeated at 6-month intervals for the first 3 years of life and annually, thereafter, through adolescence. Closer follow-up is recommended for children with cCMV-associated SNHL and tailored to the rapidity and progression of SNHL in affected children. Comprehensive eye exams are recommended in the newborn period, with close follow-up as needed and developmental assessments with appropriate interventions as necessary.

### 2.6. Conclusions

In conclusion, antiviral (valacyclovir) treatment during pregnancy to decrease maternal to fetal HCMV transmission, while promising, is limited by the fact that the studies are only applicable to primary maternal infections and require universal prenatal HCMV screening for identifying pregnant women with primary HCMV infections. Due to these limitations, anti-HCMV antiviral treatment is not currently routinely offered during pregnancy. Similarly, ganciclovir/valganciclovir treatment for cCMV has only shown to be effective in newborns with symptomatic cCMV with or without CNS involvement and needs to be initiated within the first month of life for improved audiological and neurodevelopmental outcomes. Irrespective of the antiviral interventions used in the pregnancy and newborn periods, children with cCMV will need close audiological and developmental evaluations with timely interventions to improve overall outcomes.

## Figures and Tables

**Table 1 viruses-15-02116-t001:** Randomized clinical trials with evidence for antiviral treatment efficacy in HCMV maternal and congenital infections.

Study	Study Objective	Sample Size	Country	Conclusions
**Maternal HCMV Infections**				
Shahar-Nissan K, et al. Valaciclovir to prevent vertical transmission of cytomegalovirus after maternal primary infection during pregnancy: a randomized, double-blind, placebo-controlled trial. Lancet. 12 September 2020; 396(10253): 779–785. [16]	To determine the efficacy of oral valacyclovir in preventing maternal–fetal HCMV transmission in pregnant women with primary maternal HCMV infections	90 pregnant women	Israel	Following 6 weeks of treatment with valacyclovir, fewer women with primary HCMV infections who received treatment tested positive for HCMV on amniocentesis at 21–22 weeks of pregnancy (11% vs. 30%; *p* = 0.27), particularly for women with first-trimester infections (11% vs. 48%; *p* = 0.02).
**Congenital HCMV infection (cCMV)**				
Kimberlin DW, et al. Effect of ganciclovir therapy on hearing in symptomatic congenital cytomegalovirus disease involving the central nervous system: a randomized, controlled trial. J Pediatr. July 2003; 143(1): 16–25. [48]	To assess audiological outcomes in neonates with symptomatic cCMV involving the central nervous system randomly assigned to receive intravenous ganciclovir vs. no treatment (placebo).	100 infants (Evaluable data from 42)	United States of America	Ganciclovir recipients vs. the control group showed some improvement in or maintained hearing status at 6 months (84% vs. 59%, respectively; *p* = 0.6).
Kimberlin DW, et al. Valganciclovir for symptomatic congenital cytomegalovirus disease. N Engl J Med. March 2015; 372(10): 933–943. [51]	To determine audiological and neurodevelopmental outcomes in neonates with symptomatic cCMV with valganciclovir for 6 weeks vs. 6 months.	96 infants	United States of America	Improved hearing outcomes at 12 months in infants treated for 6 months vs. those treated for 6 weeks (73% vs. 57%, *p* = 0.01). At 24 months, sustained benefit for hearing outcomes (77% vs. 64%, *p* = 0.04) and neurodevelopmental scores (*p* = 0.004).

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
