# Peer review of "Antiviral Treatment of Maternal and Congenital Cytomegalovirus (CMV) Infections"

_viruses, 2023, doi:10.3390/v15102116_

Round 1
Reviewer 1 Report
In this work, Drs. Pinninti and Boppana, provide an authoritative update on approaches to treatment of maternal and congenital human cytomegalovirus (cCMV) infections. Congenital infections with CMV are not rare. In addition to their impact on individuals and families, they collectively impose a significant societal burden. Studies conducted internationally over the past two decades have demonstrated the value of blocking CMV replication to reduce the burden of cCMV, but the practical challenges of doing this on a wide scale make it clear that development of an effective CMV vaccine remains a high priority. The well-written article is timely and will be widely read.
Suggestions for improving the manuscript:
1. Razonable et al. (coauthored by Drs. Pinninti and Boppana) provide a reasonably comprehensive review of diagnostic issues connected to cCMV (PMID 32134488) that should be cited here.
2. The paper is presented as a series of study summaries. I have two suggestions that would make the story easier to digest:
a. Create a table that summarizes the various major studies (purpose, number of patients, country, main conclusions, reference). With such a table in hand, the text could be shortened, without losing content.
b. Add a diagram illustrating the several kinds of relationships between maternal infections and cCMV.
3. Line 229: The comment about ease of collection and contamination seems to be an opinion based on extensive local experience with one of the two approaches. The sophisticated and successful national approach to urine collection in Japan warrants respect.
Minor points
a. Given the broad readership and scope of Viruses, be explicit that the “CMV” being discussed is human cytomegalovirus.
b. It seems helpful to include information about the countries where the various studies took place, in part to make it clear that this is a problem of international scope and effort. This information can be included in the suggested table.
c. Do the authors draw any conclusions from the information presented in lines 307-313?
d. Line 321. Was the neutropenia chronic or transient?
Suggested edits:
line 78: “studies”
lines 117-118: “that the majority”
Line 118: “cost analysis” to “cost effectiveness”
Line 186. “the requirement for” to “would require widespread application of”
line 189: “and of” to “and”
Line 196: “While most of the burden attributable to cCMV is due to non-primary”
Lines 197-198: “has been shown to be” to “is”
Line 227: “The high levels of CMV DNA in the saliva and urine of enable use of either specimen to identify”
Line 293: “revealed” to “determined”
Lines 299-300. “at 12 months, the hearing of more of the children treated for six months improved than for those treated for only 6 weeks”
Line 305. “recommended” to “include”
Line 311. “reported” to “showed”
Line 322. “who received valganciclovir for one year and were then followed for an average of …”
Reviewer 2 Report
This review by Pinninti and Boppana summarizes our current knowledge concerning the presently used strategies to prevent/treat maternal or congenital CMV infection. While most parts comprehensively describe the literature I feel that the paragraph focusing on the use of HIG for prevention of prenatal infection lacks information. It would help the reader to include a discussion on potential reasons why HIG treatment might have failed so far.
Author Response
Reviewer #2 Comments and Responses
This review by Pinninti and Boppana summarizes our current knowledge concerning the presently used strategies to prevent/treat maternal or congenital CMV infection. While most parts comprehensively describe the literature I feel that the paragraph focusing on the use of HIG for prevention of prenatal infection lacks information. It would help the reader to include a discussion on potential reasons why HIG treatment might have failed so far.
We would like to thank the reviewer for feedback and comments. We apologize for the brief descriptions of few of the sections in this review, specifically, behavioral and hygiene intervention strategies, hyperimmune globulin and HCMV vaccines. Due to journal requirements to primary focus on anti-viral treatment of maternal and congenital CMV infections and space constraints, the limitations of HIG in pregnancy are not discussed in detail.